# Enrofloxacin Dose Optimization for the Treatment of Colibacillosis in Broiler Chickens Using a Drinking Behaviour Pharmacokinetic Model

**DOI:** 10.3390/antibiotics10050604

**Published:** 2021-05-19

**Authors:** Robin Temmerman, Ludovic Pelligand, Wim Schelstraete, Gunther Antonissen, An Garmyn, Mathias Devreese

**Affiliations:** 1Department of Pharmacology, Toxicology and Biochemistry, Faculty of Veterinary Medicine, Ghent University, Salisburylaan 133, 9820 Merelbeke, Belgium; robin.temmerman@ugent.be (R.T.); wim.schelstraete@huvepharma.com (W.S.); gunther.antonissen@ugent.be (G.A.); 2Department of Clinical Sciences and Services, Royal Veterinary College, University of London, Hawkshead Lane, Hatfield AL9 7TA, UK; lpelligand@rvc.ac.uk; 3Department of Pathology, Bacteriology and Poultry Diseases, Faculty of Veterinary Medicine, Ghent University, Salisburylaan 133, 9820 Merelbeke, Belgium; an.garmyn@ugent.be

**Keywords:** colibacillosis, dose optimization, population pharmacokinetics

## Abstract

Enrofloxacin is frequently administered via drinking water for the treatment of colibacillosis in broiler chickens. However, the EMA/CVMP has urged to re-evaluate historically approved doses, especially for antimicrobials administered via drinking water. In response, the objectives of this study were two-fold. First, to evaluate the pharmacokinetics (PK) of enrofloxacin following IV, PO and drinking water administration. Second, to predict the efficacy of a range of doses in the drinking water for the treatment of APEC infections. For the first objective, PK parameters were estimated by fitting a one-compartmental model with a zero-order IV infusion and an oral absorption lag function to the simultaneously modelled IV and PO data. After fixing these parameter values, a drinking behaviour pharmacokinetic (DBPK) model was developed for the description and prediction of drinking water PK profiles by adding three model improvements (different diurnal and nocturnal drinking rates, inter-animal variability in water consumption and taking account of dose non-proportionality). The subsequent simulations and probability of target attainment (PTA) analysis predicted that a dose of 12.5 mg/kg/24 h is efficacious in treating colibacillosis with an MIC up to 0.125 μg/mL (ECOFF), whereas the currently registered dose (10 mg/kg/24 h) reaches a PTA of 66% at ECOFF.

## 1. Introduction

Enrofloxacin is a second-generation fluoroquinolone with a high potency against Gram-negative aerobic bacteria, exclusively developed for veterinary use [1]. Enrofloxacin is still an important part in the veterinarian’s armamentarium for the treatment of avian colibacillosis in many parts of the world [2,3]. Colibacillosis refers to any localized or systemic infection caused by avian pathogenic *Escherichia coli* (APEC) [4,5]. However, the use of fluoroquinolones is becoming increasingly restricted through legislation or even prohibited in some countries (like the USA) as this antimicrobial class is of critical importance for human medicine [6].

The European Medicines Agency’s Committee for Veterinary Medicinal Products (EMA/CVMP) stated in 2018 that enrofloxacin should no longer be used for the treatment of *E. coli* in broilers and turkeys via drinking water. The reason for this was because market authorization holders failed to provide pharmacokinetic/pharmacodynamic (PK/PD) data to justify the approved doses [7]. More recently, in 2019, EMA/CVMP released a draft stating the urgency to optimize the dosage regimens of antimicrobial drugs currently employed in veterinary medicine, especially those administered via drinking water, using modelling/simulation and probability of target attainment (PTA) analysis [8]. 

As is demonstrated above, there is a clear need for optimizing the currently approved dosage regimen of enrofloxacin for APEC infections. The current veterinary dosage regimens are determined during drug development via dose-ranging (titration) and dose confirmation studies using pre-defined doses, treating the dose–effect relationship as a “black box”, since no or little information is gathered on the PK/PD processes. Subsequently, doses estimated via this approach are susceptible to bias, misinterpretation and are usually suboptimal [9,10]. The optimization of dosage regimens of antimicrobial drugs using PK/PD modelling is accomplished by selecting the dosage regimen that ensures that 90% or 95% of the patient population (PTA) achieves a specific value of a PK/PD target index [11]. For fluoroquinolones, the index associated with clinical efficacy against Gram-negative bacteria is the ratio AUC(24 h)/MIC ≥ 125 h [12,13,14,15].

Enrofloxacin is generally administered via drinking water in broilers [16]. Administration via this route adds extra variability to the PK process, which is related to the individual’s water (or feed) intake. However, there are limited examples available illustrating the compellingly large variation in drinking or feeding behaviour and their impact on exposure [17,18,19]. Chicken drinking behaviour is also influenced by fluctuations in temperature and the type of drinker available [20]. When optimizing doses, this variability in dose uptake has to be taken into consideration. Studies concerning the modelling of variable drug input are scant. In a study regarding patient compliance in human medicine, dose uptake variability was modelled by creating a stochastic drug input in the PK model [21,22]. The same authors investigated oxytetracycline administered in pig feed and developed the new concept of a feeding behaviour pharmacokinetic (FBPK) model to account for the feeding behaviour-induced PK variability [23]. Studies about PK/PD-modelling of drinking water administration while taking drinking behavioural variability into account have so far, to the best of the author’s knowledge, not been published.

Therefore the aims of this study were two-fold. First, we developed an innovative drinking behaviour PK (DBPK) model of enrofloxacin in broilers, by evaluating the PK of enrofloxacin following IV, PO and drinking water administration, and taking inter- and intra-individual sources of variability into account, including the variability in drinking water uptake. Second, we performed simulations with the previously established DBPK model to compute the minimal dose to achieve the PK/PD target (AUC (24 h)/MIC = 125 h) in 90% of individuals at the MIC corresponding to the epidemiological cut-off (ECOFF) of *E. coli*, which is 0.125 µg/mL.

## 2. Results

Figure 1 gives an overview of the PK modelling process and the final DBPK model. A one-compartment model with zero-order IV infusion (IV delay = ±3 min), and first-order absorption and elimination with a lag time for the oral absorption process (Tlag = ±2.4 min) best fitted the simultaneously modelled IV and PO data. After fixing the parameters, the drinking water data were added to the model and two additional parameters were estimated, namely, Drink and dVddose.

The typical values (tv) of the primary structural parameters of the model, their associated standard error (SE), coefficient of variation (CV%), 95% confidence intervals and between-subject variability (BSV) for the final model are presented Table 1. The secondary parameters can be found the Supplementary Files (S1). The precision of most model parameters was considered acceptable (CV% < 20). BSV was 11%, 22% and 100% for Cl, V and Ka, respectively. Therefore, most of the BSV resided in the absorption process rather than clearance and volumes. 

The typical value of the parameter describing water uptake, Drink, was approximately 194 mL/kg/24 h. This corresponds to a mean water consumption of around 316 mL/24 h per bird (mean weight 1.63 kg). The parameter was estimated with good precision (5.8 CV% and the 95% confidence interval between 182.5 and 205.3). The associated BSV was 24%. 

The predicted quantiles (10%, 50%, and 90%) of the stratified visual predictive checks (VPCs) were close to the corresponding observed quantiles, indicating a good model fit (Figure 2). The adequacy of the model can be further verified from the IPRED and PRED vs. observed concentrations and CWRES vs. time goodness-of-fit plots (Appendix A).

There was good correlation (Figure 3) between the observed water consumption and the water consumption estimated by the model (R^2^ = 0.802). 

The variance–covariance matrix and shrinkage are given in Table 2. Low levels of η-shrinkage (<30%) were indicative of appropriate data richness, which in turn meant precise estimation of the random effect parameters of the model.

Using MCS and the previously finalized population DBPK model, virtual enrofloxacin PK profiles were generated for 50 replicate sets of 15 animals per dose, over an 80 h period. The following nine doses were tested: 2.5, 5, 7.5, 10, 12.5, 15, 20, 30 and 50 mg/kg/24 h. The IPREDs were simulated, taking into account the structural parameters (θ) and BSV (η), but not the residual error (σ). These curves were then used to compute the PTA (%) of the PK/PD-breakpoint fAUC_(48–72h)_/MIC for the different doses and several MICs (0.008–0.016–0.032–0.064–0.125–0.25–0.5–1–2 µg/mL), taking only the free concentration into account (80% of the total concentration). The results of the PTA analysis are displayed in Figure 4.

Based on the PK/PD index of AUC_(48–72h)_/MIC of 125 h, which is equivalent to fAUC_(48–72h)_/MIC of 100 h, the registered dose of enrofloxacin (10 mg/kg/24 h) is predicted to be effective for covering the APEC strains up to an MIC of 0.064 µg/mL (PTA 100%), but only achieves a PTA of 66% at 0.125 µg/mL, which corresponds to the epidemiological cut-off (ECOFF). The ECOFF categorizes strains into wild-type and non-wild-type, based on the absence or presence of an acquired or mutational resistance mechanism. For a dose of 12.5 mg/kg/24 h or higher, the PTA was 90% for an MIC of 0.125 µg/mL, which could be established as the PK/PD cut-off. A selective overview of the PTA analysis is given in the Appendix A.

## 3. Discussion

The dose optimisation of enrofloxacin for the treatment of colibacillosis in broiler chickens improves the One Health sustainability of antimicrobial agents for three reasons. First, inadequate dosing and subsequent exposure can lead to treatment failure and resistance development in the target pathogen [24]. Second, APEC strains have zoonotic potential [25,26,27] and can lead to human infections, which are more difficult to treat because of the increased resistance to antimicrobial drugs. Finally, resistance genes can be exchanged via horizontal gene transfer between the bacteria of animals and humans, where the environment can have a facilitating role [24].

The first goal of this study was to develop a DBPK model of enrofloxacin administered in broilers, describing the absorption and disposition of the drug administered via drinking water while taking the inter-animal variability (IIV) into account.

Modelling this modality of administration allows clear extrapolation from the in silico models to the veterinary poultry practice, where individual IV and PO administration is almost non-existent.

The estimated PK parameters were chiefly in alignment with the established literature. A typical clearance of 485 mL/kg/h was estimated, which is an approximately two-fold increase from the values reported by two previous studies [28,29]. However, one study reported a higher mean total body clearance after IV administration to broilers, namely, 10.3 mL/(min*kg) (i.e., 618 mL/(h*kg)) [30]. Similar to the relationship between the clearance estimated in this study and the reported clearance values in the literature, the mean estimated volume of distribution (4672 mL/kg) fell in between the mean results estimated by other studies, which ranged from 1.98 L/kg to 5 L/kg [28,29,30]. Concerning bioavailability (F), there is again some variability in the published studies, with values ranging from 60% [31] to 80% [28] and 90% [30]. Our estimation of F (78.80%) falls yet again amid this range. 

The kinetics of enrofloxacin administered via drinking water were modelled as a zero-order rate infusion, differentiating between diurnal and nocturnal infusion rates and taking inter-animal variability into account. One limitation of this approach is that the analysis and estimation of the drinking behaviour was performed the day before the administration of enrofloxacin and on a limited sample of 15 animals, who were later exposed to the same dose of 20 mg/kg/24 h. For a more accurate analysis, the drinking water uptake should be evaluated during the enrofloxacin administration and on a large number of animals exposed to different doses. This would then allow for clear extrapolation between the measured uptake and the exposure (expressed as AUC), and the appraisal of possible dose-dependent increases or decreases in the water uptake. Nevertheless, a good correlation was found between both the drinking data before administration and the subsequent exposure after administration, and the observed and estimated water consumption. 

The effect of temperature fluctuations on drinking behaviour was not assessed in this study. A heuristic in poultry practice states that per degree Fahrenheit increase above the ambient stable temperature, the birds will drink approximately 7% more. However, many studies show that drinking water uptake is only significantly increased in periods of heat stress [20,32,33,34,35]. Heat stress occurs when the ambient temperature is above the thermal comfort zone of the birds, which for adult broilers is generally above 30 °C [36,37,38]. In commercial broiler farms, the temperature is strictly controlled through ventilation. Therefore, major temperature deviations will be mitigated most of the time. There are exceptions of course ((sub)tropical climates and excessively hot summers), but these are of less importance in the target region of Western Europe.

Moreover, temperature-mediated alterations in the drinking behaviour of the flock will not impact the administered dosage, since the concentration of the antimicrobial agent added to the drinking water is dependent on the water consumption of the previous 24 h. This can be appreciated from Equation (1). However, for many antimicrobial agents and products, this dose calculation is not stated on the leaflet. Therefore, it is possible that concentration correction of the drug based on the water intake is not always done in practice. However, it can be considered good veterinary/agricultural practice (GVP/GAP) to always determine the drug concentration that is needed to be added to the drinking water to attain a certain dose based on the average water consumption of the flock.

In this study, the potential shift in water uptake was corrected for in the dose calculation. For this reason, an ambient temperature covariate that could influence the average water (and drug) uptake at different temperatures was not included in the DBPK model. However, future studies should be conducted to evaluate the impact of temperature deviations on drinking behaviour and the subsequent absorption, disposition and efficacy of antimicrobial agents in more detail.

Of note was the bird who drank notably less (and resulted in a lower AUC) than the other broiler chickens (Figure 6, grey dot). While evaluating the video recording, this animal moved around less frequently and appeared debilitated. This anecdotal observation highlights the importance of early treatment intervention (metaphylaxis) during infections, since these have an impact on the drinking and feeding behaviour of animals, resulting in a lower exposure when very diseased. 

Several doses were administered to the birds via drinking water, ranging from 2.5 to 20 mg/kg/24 h. A nonlinear relationship between exposure (AUC) and increasing doses was observed. Several approaches were tested to capture the nonlinearity.

First, the inclusion of (intestinal) efflux mechanisms was modelled, since enrofloxacin is a well-known p-glycoprotein (p-gp) substrate [39]. Other tested mechanisms were saturable Michaelis–Menten absorption and Imax-driven decrease (inhibitory Emax model) in absorption. None of these mechanisms significantly improved the model.

Notably, when the nonlinear mechanism was enclosed in the drinking behaviour (see Equation (6), parameter dVddose), the model fit improved significantly. One explanation for a decrease in the water uptake associated with higher doses could be an increased perception of the taste of enrofloxacin. However, this effect was not seen in a study where they administered a dose of 50 mg/kg/24 h to the birds [40]. Another explanation could be the increase in variability of the drug concentrations drank by the birds at higher doses, where birds drinking more or less than average have greater effects on exposure since the medicated water is more concentrated. 

In the second part of this study, the PTA of different doses was determined for the PK/PD index associated with the clinical efficacy of fluoroquinolones against Gram-negative bacteria (AUC/MIC ≥ 125 h). This approach is similar to the methodology stipulated by VetCAST for the determination of a veterinary clinical breakpoint [41,42]. The registered dose of enrofloxacin (10 mg/kg/24 h) administered via drinking water is effective in treating APEC strains with MIC values equal to or lower than 0.064 µg/mL (100% PTA). Susceptible strains with an MIC value equal to the ECOFF (0.125 µg/mL) are covered by a dose of 12.5 mg/kg/24 h (90% PTA). In contrast, susceptible strains with an MIC equalling the CLSI clinical breakpoint of susceptibility (0.25 µg/mL) are only covered by the highest investigated dose of 50 mg/kg/24 h (90% PTA). 

The discrepancy between the ECOFF and CLSI clinical breakpoint of susceptibility is of major importance, since the strains that are still deemed susceptible in antimicrobial susceptibility testing (but with an MIC above the ECOFF) are difficult to treat because they require an approximately five-fold increase from the currently approved dose. This raises concerns of the validity of the currently established clinical breakpoint for enrofloxacin and *Escherichia coli*. According to VetCAST, veterinary clinical breakpoints are based on the following three separate components: an ECOFF, a PK/PD breakpoint and a clinical cut-off based on clinical data [42]. However, the latter is generally not available in veterinary medicine. The PK/PD breakpoint is the highest possible MIC for which a given percentage of animals in the target population achieves a critical value for the selected PK/PD index using the registered dose. In this study, the PK/PD cut-off was 0.064 µg/mL (just one dilution below ECOFF), but the PTA was still high (66%) at ECOFF for the registered dose. 

Using the conservative PK/PD index of 125 h (and breakpoint values in general) has limitations, however. The PK/PD indices are summary endpoints and do not provide detailed information about the time course of the PK and PD processes [43]. Instead, the development of mechanistic mathematical models characterizing the full time-course of the PK/PD mechanisms provides a more accurate description of antimicrobial activity [44,45,46,47]. This includes quantitative analysis of time–kill curve (TKC) data. This approach has already been introduced in veterinary medicine [48,49]. In chickens, there are indications that the AUC_(0–24h)_/MIC for clinical effectiveness could be lower than 125 h, which would impact the choice of optimal dose [50,51]. Therefore, it is important to conduct further research to investigate the dynamics between the concentrations of enrofloxacin and *E. coli* concentrations using TKC analysis, and to develop (semi-)mechanistic PK/PD models to predict more accurate PK/PD indices that are specific with regards to the drug, the pathogen and the animal species.

## 4. Materials and Methods

### 4.1. Animal Trials

The study was approved by the ethical committee of the Faculties of Veterinary Medicine and Bioscience Engineering of Ghent University (EC2017/94). Care and treatment of the birds were in full compliance with the national and European legislation concerning animal welfare [52,53]. Throughout the studies, feed and drinking water were provided ad libitum. 

The PK of enrofloxacin were evaluated in 4 different animal trials (intravenous administration (IV), oral gavage administration (PO) richly sampled, PO sparsely sampled and drinking water administration). Temperature in the stables was set depending on the age of the birds, as described in international guidelines [54]. In the richly sampled IV and PO trials, 8 birds (4 weeks of age) received 10 mg/kg enrofloxacin intravenously (wing vein) and to another 8 birds, the same dose of enrofloxacin was administered via oral gavage. Blood samples were collected (from leg and/or wing veins) at 0 (prior to administration), 5, 10, 20, 30, 45, 60 min and 1.5, 2, 3, 4, 6, 8, 10 and 24 h post administration (p.a.). 

Regarding the sparsely sampled PO trial, 120 animals were allocated to 12 different treatment groups (*n* = 10). All birds of each group received an oral bolus enrofloxacin of 10 mg/kg. Groups 1 to 3 were treated when the animals were 27 days of age, groups 4–6 on 29 days of age, groups 7–9 on 34 days of age and lastly groups 10–12 on 36 days of age. The sample points (ranging from 5 min to 32 h p.a.) were randomly assigned to the different treatment groups. Each animal was sampled 5 times, resulting in 600 sample points in total. An overview of the sampling design is given in the Appendix A.

In the last animal trial (enrofloxacin medicated drinking water), broiler chickens (5 weeks old) were randomly assigned to 5 treatment groups, each treatment group receiving a different dose (2.5, 5, 10, 15 and 20 mg/kg/24 h). The temperature in the stables was set and controlled at 20 °C, conforming to international guidelines [54]. The photoperiod was 16 h (7 a.m.–11 p.m.) and the scotoperiod was subsequently 8 h. One treatment group consisted of 3 pens, with 5 animals per pen, resulting in a total number of 15 animals. 

The amount of enrofloxacin (mL of Baytril^®^ 10% oral solution, Bayer, Diegem, Belgium) that needed to be added in the drinking water was calculated using the following formula (Equation (1)):(1)$$\frac{Dose\;\left(\frac{mg}{kg}\right)*average\;BW\;\left(\mathrm{kg}\right)*number\;of\;animals\;per\;group*amount\;of\;water\left(L\right)}{Average\;water\;uptake\;per\;group\;in\;24\;h\;\left(L\right)}$$

The average water uptake of the broiler chickens was determined by weighing the drinking bells of 5 randomly selected groups at a specific time point and 24 h after. The average body weight was calculated by taking the average of all the birds present in those 5 groups (*n* = 25). 

After an acclimatization period (1 week), the drinking water was medicated with enrofloxacin for 3 consecutive days. The drug was administered at 9 a.m. A first blood sample was taken 8–9 h later (3 p.m.). For the next 2 days, blood samples were taken shortly after the light was turned on in the stable but before administration (7 a.m.) and then 8–9 h later around 3 p.m. The last blood sample was collected 79 h p.a. This resulted in 6 samples per animal (7, 23, 31, 46, 55 and 79 h p.a.). An overview of the experimental design of the drinking water administration trial is shown in the following figure (Figure 5).

Additionally, 15 animals were filmed during the drinking water administration trial (the day before drinking water administration) to evaluate the correlation between drinking water consumption and enrofloxacin exposure on the subsequent day. 

After the experiments, animals were euthanized with pentobarbital IV (sodium pentobarbital 20%, Kela, Hoogstraten, Belgium). The sex of the animals was determined post mortem.

### 4.2. Quantification of Enrofloxacin in Plasma

The technique used in this study for the analysis of enrofloxacin in plasma was based on the method described by Devreese et al. [1], with slight modifications. Concisely, 250 µL of plasma sample was spiked with 25 µL of internal standard solution (1 µg/mL) and vortex mixed (±15 s). Next, 3 mL of ethyl acetate was added to the samples and these were extracted on a roller mixer for ±15 min (liquid–liquid extraction). After mixing, the samples were centrifuged (2851× *g*, 10 min, 4 °C). Thereafter, the organic layer was transferred to another tube and evaporated under a gentle nitrogen (N2) stream (45 ± 5 °C). Reconstitution of the dry residue was achieved in 250 µL of ultrapure H2O. Finally, the sample was transferred to an autosampler vial after filtering through a 0.20 µm nylon filter (Merck Millipore, Overijse, Belgium) and the aliquot (10 µL) was injected into the LC–MS/MS system.

Chromatographic separation was achieved on a Zorbax Eclipse Plus column (reversed phase C18, 100 mm × 3.0 mm i.d., dp: 3.5 µm) in combination with a guard column of the same type (13 mm × 3.0 mm i.d., dp: 3.5 µm) (Agilent Technologies, Diegem, Belgium) using a gradient elution programme consisting of 2 mobile phases (A and C). Mobile phase A consisted of 0.1% glacial acetic acid in ultrapure H2O and mobile phase C was 100% ACN. The following gradient elution program was applied: 0–3 min (80% A, 20% C), 3–3.5 min (linear gradient to 50% A), 3.5–5 min (50% A, 50% C), 5–5.5 min (linear gradient to 80% A) and 5.5–10 min (80% A, 20% C). 

Flow rate was set at 0.3 mL/min. After separation, the LC eluent was coupled to a Waters Quattro Ultima^®^ (Asse, Belgium) triple quadrupole mass spectrometer with ion source heated electrospray ionization (h-ESI) operating in positive ionization mode. Acquisition was performed in selected reaction monitoring (SRM) mode. For enrofloxacin and the IS, the following transitions were followed (*quantification ion): ENR: m/z 360.0 > 316.1, 242.3* and ENR-d5: m/z 365.0 > 321.1. The LC–MS/MS analytical methods were validated for ENR using matrix-matched calibrator and quality control samples, based on blank plasma of untreated broiler chickens.

The limit of quantification (LOQ) was 50 ng/mL. The LC–MS/MS analyses were conducted in accordance with international guidelines [55,56,57]. 

### 4.3. PK/PD Modelling and Simulation 

The pharmacokinetic data analyses were conducted using Phoenix^®^ 8.2 (Certara, Princeton, NJ, USA). The PK profiles of the richly sampled PO and IV datasets were analysed simultaneously with the sparsely sampled PO and drinking water administration datasets using NLME, as previously described [10]. Estimation of the PK parameters and their associated variability was performed by minimizing the objective function value (OBV), the negative 2 log-likelihood (-2LL), using Laplacian maximum likelihood estimation.

Data below the LOQ (BLQ) were excluded (M1 method) with negligible bias for parameter estimation, since these constituted approximately 0.02% of the combined dataset [41,58,59,60]. 

The PK data was modelled sequentially. In the first step, the PK data of the birds administered IV (*n* = 8) together with the data of the oral gavage routes (8 densely and 120 sparsely sampled birds) were modelled simultaneously.

Standard goodness-of-fit diagnostics, including individual (and population) vs. concentration plots and the conditional distributions of weighted residuals (CWRES) were used to assess the performance of the candidate models. 

Two concurrent, nested models were compared using the likelihood ratio test (LRT). For the addition of one parameter, given a significance level of 0.05 (type I error), the critical value of the χ^2^ distribution is 3.84. Therefore, a decrease of 3.84 in the objective function value (OBV) with an additional model parameter is significant. A one-compartment model was chosen to fit the data. The basic mathematical PK model can be written as follows (Equation (2)):(2)$$\begin{array}{l} Y=f\left(\theta_{i},\;Time\right)\times \left(1+\varepsilon_{1}\right)+\varepsilon_{2}\\ \theta =\left[\begin{array}{c} \begin{array}{c} \begin{array}{c} F_{i}\\ k_{ai}\\ Cl_{i}\\ V_{i}\\ IV\_delay_{i}\\ Tlag_{i} \end{array} \end{array} \end{array}\right]=\left[\begin{array}{c} \begin{array}{c} invlogit\left(logit\;\left(tvF\right)\right)\\ tvk_{a}\times \exp\left(\eta_{k_{ai}}\right)\\ tvCl\times \exp\left(\eta_{Cl_{i}}\right)\\ tvV\times \exp\left(\eta_{V_{i}}\right)\\ tvIV\_delay\\ tvTlag \end{array} \end{array}\right] \end{array}$$
where *Y* are the observed plasma concentrations, *f*(*θ_i_*, *Time*) represents the structural model and the (1+ $\varepsilon_{1}$) and the $\varepsilon_{2}$ terms characterize the residual error model. A combined error model was chosen, consisting of a multiplicative term $\varepsilon_{1}$ and an additive term $\varepsilon_{2}$, having both a mean of 0 and a standard deviation of σ_1_ and σ_2,_ respectively. 

The individual deviations from typical values (*θ*) of the structural parameters, ηka, ηCl and ηV, describing the biological between-subject variability (BSV), were coded as log-normally distributed. The distribution of η in the log-domain has a mean of 0 and a variance of *ω*^2^. The bioavailability parameter *F* was logit transformed to bound its estimated value between 0 and 1 and was considered fixed. An IV disposition delay (IV_delay) and a lag in oral absorption (Tlag) parameter were estimated (typical values only). The full variance–covariance matrix was selected. The reported variance values (*ω*^2^) of the BSV were transformed to represent coefficients of variation (CV%) with the following equation (Equation (3)) [41,49]:(3)$$CV\;\left(\%\right)=100\times \sqrt{\exp\left(\omega^{2}\right)-1}.$$

Eta (*η*)—shrinkage was calculated to evaluate the quality of the empirical Bayes estimates (EBEs) using the following equation (Equation (4)) [61]:(4)$$Eta-shrinkage=1-\frac{SD\left(EBE_{\eta }\right)}{\omega }$$
where *ω* is the population model estimate of the variability in the SD of *η* (variability in the population) and SD (EBEη) is the SD of the individual values of the EBE’s of *η*. When *η*-shrinkage is high (non-informative and sparse data), the individual parameter estimates will deviate less from the population mean and the variance in the EBE distribution will be shrinking towards zero. Shrinkage below 30% was considered acceptable [62].

After initial estimation of the fixed and random-effect parameters from the IV and PO (rich and sparse) datasets, their typical values and variances were subsequently fixed. Thereafter, the drinking water PK data was included in the model. Essentially, group administration in drinking water is comparable to a prolonged monotonous oral administration (with varying diurnal and nocturnal rates) where chickens self-medicate depending on their drinking behaviour. As a consequence, three model improvements that are specific to the PK of group administration via the drinking water of animals were successively implemented. 

Firstly, based on the video recording analysis of 15 animals obtained during the drinking water PK experiment and the weighing of the drinking bells, it was estimated that 89.5% and 10.5% of the total daily dose was drunk during the photoperiod (16 h) and scotoperiod (8 h), respectively. Succession of day–night doses, apportioned to the water consumption in the photo- and scoto-period, respectively, were inputted as infusions in Phoenix using the ADDL function (depending on the timespan of the day/light period and the enrofloxacin concentration of the medicated drinking water). 

Secondly, there was evidence of variability in water uptake between the different birds. Indeed, there was a high correlation (R^2^ = 0.813, Figure 6) between the amount of water drunk by 15 filmed individual broiler chickens on the day before dosing (photoperiod) and their subsequent corresponding AUC (0–72 h) after administration of 20 mg/kg/24 h enrofloxacin (calculated by non-compartmental analysis). Note the one outlier on the left (grey dot) who drank very little, which is examined further in the discussion.

To account for the variability in drinking water uptake between animals, an additional parameter Drink (mL/kg/24 h) with associated BSV (ηDrink) was introduced as a factor influencing bioavailability (F) according to the following formula (Equation (5)):(5)$$F_{i}=invlogit\left(logit\;\left(tvF\right)\right)\times Drink.$$

Thirdly and finally, a lack of dose proportionality had to be implemented. Indeed, the dose proportionality between the 5 administered doses in water was refuted by evaluating the AUC (0–72 h) determined via non-compartmental analysis, using a power model. To account for the observed less than proportional exposure (AUC) to dose relationship, we included a covariate (dVddose) that decreased the tvDrink proportionally with dose, as described by the equation (Equation (6)):(6)$$Drink=tvDrink+\left(dose-mean\left(dose\right)\right)\times dVddose+\eta_{Drink}$$
where mean(dose) refers to the average of the 5 administered doses, which is 10.5 mg/kg/24 h.

The final model, which described the absorption and disposition of enrofloxacin following drinking water administration while taking variability in water uptake into account, is coined a DBPK model.

The validity (internal) of the final model was assessed by plotting stratified visual predictive checks (VPCs) for each distinct dataset (IV, PO sparse, PO rich and the different doses administered via the drinking water). The 95% confidence intervals of the 10th, 50th and 90th percentiles of the simulated PK profiles were matched with corresponding percentiles of the observed data.

### 4.4. Simulation and Probability of Target Attainment Analysis

Next, a virtual population was generated by MCS in Phoenix (50 replicates of the PK dataset containing 211 birds) with different PK profiles after drinking water supplemented with different doses. 

The free plasma concentration–time profiles were simulated for up to 80 h (with increments of 1 h), using the 5 doses that were administered in the in vivo experiment (2.5, 5, 10, 15 and 20 mg/kg/24 h) and 4 additional ones (7.5, 12.5, 30 and 50 mg/kg/24 h). Protein binding of enrofloxacin in broilers is reported to be 20% [28,31]. Several areas under the curve (0–24 h, 0–48 h, 0–72 h and 48–72 h) were calculated from the simulated curves with NCA. Summary statistics and the different quantiles (ranging from 1 to 99) were determined using the Phoenix statistical tool. The PTA of the different doses was calculated using the PK/PD-breakpoint AUC(24h)/MIC ≥ 125 h [12,13,14,15] for a range of MIC values (0.0625—2 µg/mL). A PTA of 90% was considered effective, in alignment with the PK/PD cut-off determination by the EUCAST Veterinary Subcommittee (VetCAST) [41,42]. The AUC of 48 to 72 h was chosen since it is assumed that during this time frame steady-state is achieved [63]. Since the free concentrations are investigated, the AUC(48–72 h)/MIC ≥ 125 h target can be converted into an fAUC (48–72 h)/MIC target by multiplying the former with the free fraction [64], which is approximately 0.8. This results in an fAUC (48–72 h)/MIC index of 100 h.

## 5. Conclusions

In conclusion, this study introduces the novel concept of the DBPK model and describes a novel approach for the PK modelling and dose optimization of antimicrobial agents administered via drinking water in the veterinary sector. Based on the conservative PK/PD index of AUC_(0–24h)_/MIC of 125 h, the dose of 12.5 mg/kg per day is efficacious in treating colibacillosis caused by strains without acquired resistance mechanisms (MIC = ECOFF), whereas the currently registered dose (10 mg/kg per day) reaches a PTA of 66%. However, more research is needed to develop mathematical models characterizing the full time-course of the PK/PD processes, thereby providing more accurate predictions of antimicrobial efficacy and PK/PD indices.

## Figures and Tables

**Figure 1 antibiotics-10-00604-f001:**
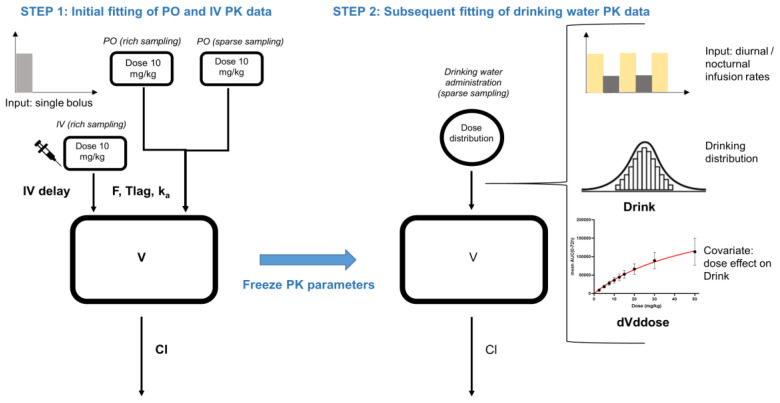
An overview of the PK modelling process. Using the data of the IV and PO single bolus administration (rich and sparsely sampled) animal trials, the parameters of the one-compartment model (Cl, V, Ka and F with an additional IV delay and Tlag) were estimated and subsequently fixed in further analyses. In order to model the drinking water administration modality and develop the DBPK model, three model improvements were implemented. First, diurnal and nocturnal infusion rates were chosen in order to mimic the drinking behaviour during the photo- and scoto-period, respectively. Second, the variability in drinking water behaviour and therefore in bioavailability was captured using an additional parameter, Drink. Third, accounting for the nonlinear relationship between dose and exposure, a covariate was introduced (dVddose) to decrease Drink and hence bioavailability proportionally with higher doses.

**Figure 2 antibiotics-10-00604-f002:**
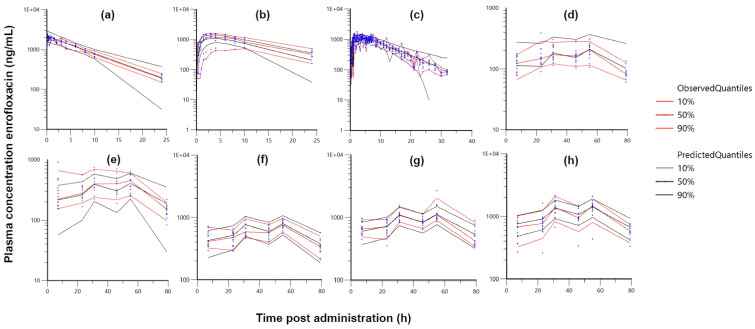
Stratified visual predictive check (VPC) of the different treatment groups ((**a**)—IV, (**b**)—PO richly sampled, (**c**)—PO sparsely sampled, (**d**)—drinking water administration 2.5 mg/kg, (**e**)—drinking water administration 5 mg/kg, (**f**)—drinking water administration 10 mg/kg, (**g**)—drinking water administration 15 mg/kg, and (**h**)—drinking water administration 20 mg/kg) with 200 replicates of each animal. Approximately 20% of the data should fall outside the plotted quantiles. Red lines: observed quantiles; black lines: predicted quantiles; blue circles: observed data.

**Figure 3 antibiotics-10-00604-f003:**
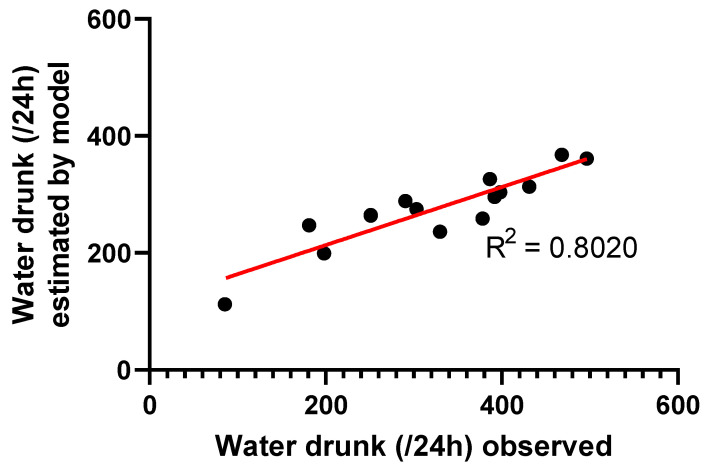
Correlation between the observed water consumption and the estimated water uptake by the model for 15 animals. The observed water drunk over 24 h was determined by adding the estimated proportion of nocturnal drinking (10.5%) to the water uptake evaluated during the photoperiod. Coefficient of determination (R^2^) = 0.8020.

**Figure 4 antibiotics-10-00604-f004:**
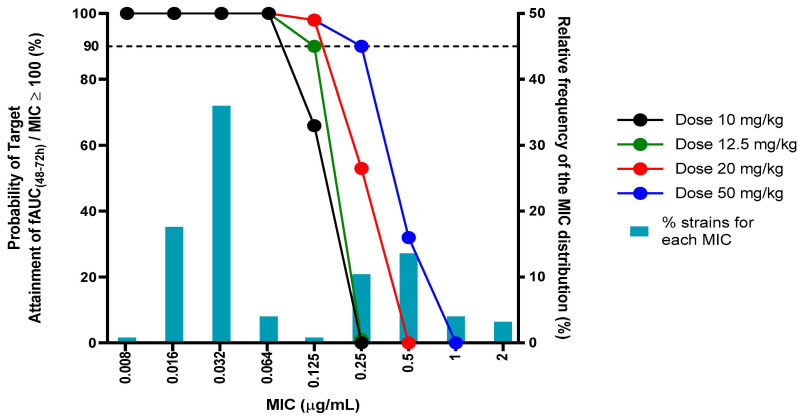
Probability of target attainment (PTA) for fAUC (48–72 h)/MIC ≥ 100 h in 75 simulated broiler chickens for four doses of enrofloxacin (10, 12.5, 20 and 50 mg/kg) administered via drinking water (five other simulated doses are omitted). The horizontal dotted line signifies the PTA of 90%. The relative frequency of the MIC distribution of APEC strains is taken from a recent study from Flanders (Belgium) [24].

**Figure 5 antibiotics-10-00604-f005:**
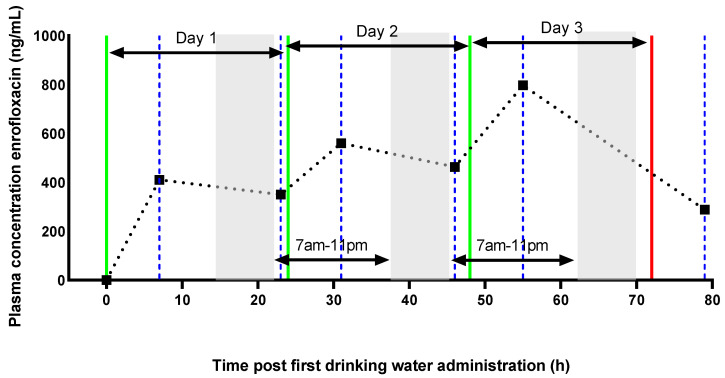
The green lines correspond with the time when enrofloxacin was introduced in the drinking water (9 a.m.) and kept for 24 h. This was done for three consecutive days. The red line indicates the cessation of enrofloxacin administration. The blue dotted lines relate to the six different sample points (7, 23, 31, 46, 55 and 79 h post administration). The grey areas indicate the periods that the light in the stable was turned off (scotoperiod, 11 p.m. to 7 a.m.), and the white areas correspond to the photoperiod. The black dotted line is an example of a PK profile of a representative broiler dosed at 10 mg/kg/24 h.

**Figure 6 antibiotics-10-00604-f006:**
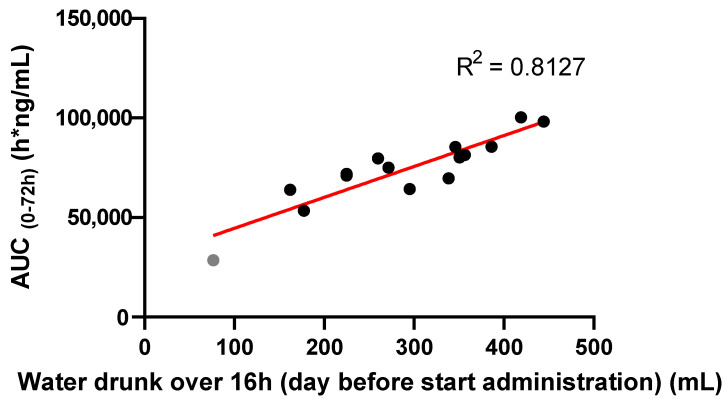
The correlation between AUC_(0–72h)_ of enrofloxacin administered via the drinking water of 15 broiler chickens dosed at 20 mg/kg and the amount of water drunk over 16 h (the filming period) per bird the day before administration. Coefficient of determination (R^2^) = 0.8127.

**Table 1 antibiotics-10-00604-t001:** Population primary (thetas) parameters of enrofloxacin in broilers obtained via sequential modelling and their associated standard error (SE), the lower and upper bounds of the 95% confidence interval (CI) and the estimated between-subject variability (BSV). First, the IV and PO data were fitted with a one-compartment model with IV delay and Tlag functions. Second, the drinking water PK data was added and the parameters Drink and dVddose were estimated.

Typical Values	Estimate	Units	SE	2.5% CI	97.5% CI	BSV (%)
Step 1: Initial modelling of IV and PO PK data						
tvCl	458.1	mL/kg/h	29.59	400.0	516.2	11
tvV	4671.9	mL/kg	339.1	4006.2	5337.7	22
tvK_a_	0.685	1/h	0.051	0.584	0.786	100
tvTlag	0.041	h	0.006	0.030	0.052	-
tvIV_delay	0.054	h	0.013	0.028	0.079	-
logit(tvF)	1.313 *		0.319	0.686	1.940	-
Step 2: Modelling of drinking water PK data						
tvDrink	193.9	mL/kg/24 h	5.827	182.4	205.3	24
dVddose	−1.849		0.461	−2.754	−0.945	-
tvCMultStdev	0.096	%	0.004	0.087	0.104	-
stdev0	116.8	ng/mL	0.748	115.3	118.3	-

* Back transformation of this value (invlogit) gives an F of 0.788.

**Table 2 antibiotics-10-00604-t002:** Estimates of the random effects variance–covariance matrix, correlation matrix and shrinkage.

Omega			
	nCl	nV	nK_a_
nCl	**0.012**		
nV	0.008	**0.048**	
nK_a_	−0.023	0.150	**0.694**
**Correlation between ETAs and shrinkage**			
nCl	**1**		
nV	0.348	**1**	
nKa	−0.247	0.823	**1**
Shrinkage	0.207	0.139	0.086

Variances (diagonal) are in bold; nCl, nV and nK_a_ are the random components of the model (ETAs). The BSVs, expressed as coefficient of variation (CV%) using Equation (2), were 10.9, 22.1 and 100.1% for nCl, nV and nKa, respectively. The variance, CV% and shrinkage of nDrink (the random component of tvDrink, not included in the variance–covariance matrix) were 1542.23, 24% and 15%, respectively.

## Data Availability

The datasets analysed during the current study are available from the corresponding author on reasonable request.

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
