# Peer review of "Enrofloxacin Dose Optimization for the Treatment of Colibacillosis in Broiler Chickens Using a Drinking Behaviour Pharmacokinetic Model"

_antibiotics, 2021, doi:10.3390/antibiotics10050604_

Round 1

Reviewer 1 Report

The article of Temmerman and coworkers, analyzes the issue regarding the enrofloxacin administration, in chicken broilers in the treatment of colibacillosis syndromes. The authors analyze in a commendable way the pharmacokinetics of that antibiotics after administration through IV, PO, and drinking water. Then, they developed an interesting model to estimate the description and predictions of drinking water PK profiles. The final simulation of PK for enrofloxacin uptake by drinking water is notable. The paper can be accepted after a minor revision of the text,

Minor comment:

line 99, There is a repetition of "was estimated"

In figure 4, there are some black square that hides the axis labels

The references list should be checked and standardize

Author Response

Thank you very much for your comments and feedback. We have addressed the issues that you raised.

Minor comment:

line 99, There is a repetition of "was estimated"

Line 99: the duplication of “was estimated” is removed accordingly.

In figure 4, there are some black square that hides the axis labels

We do not have this issue with figure 4. This will be checked again during the final check.

The references list should be checked and standardized

The reference list is now updated and standardized. This will be checked again during the final check.

Reviewer 2 Report

This is a manuscript regarding the PK/pd of enrofloxacin administered administered to chickens via drinking water.  This is a product removed and now prohibited from the US market and being re-evaluated by the EMA.

While in general this is an interesting paper, I find a MAJOR flaw with the consideration of the dosing.  Water consumption in chickens is very temperature dependent.  The authors evaluated only 15 birds over a very short time period at a fixed temperature for there modeling exercised.  Whithout greater consideration for a wider range of temperature fluctuations which would have a greater effect on drug intake, both more and less, which effect not just efficacy, but also tissue residues, the interpretation of their results is limited, at best.  Prediction of an optimized dose is left to no more than guesswork.  As written I cannot support publication of this work without better consideration of temperature on water intake.

Author Response

We thank the reviewer for his feedback. We have tried to address the issues raised to the best of our ability.

The reviewer rightly mentions the impact of temperature fluctuations on the drinking behaviour of chickens. A heuristic in poultry practice states that per degree Fahrenheit increase above the ambient stable temperature, the birds will drink approximately 7% more. However, many studies show that drinking water uptake is only significantly increased in periods of heat stress (Bruno et al., 2011; Donkoh, 1989; May et al., 1992; May et al., 1997; Mack et al., 2013). Heat stress occurs when the ambient temperature is above the thermal comfort zone of the birds, which for adult broilers is generally above 30°C (Curtis, 1983; Mascarenhas et al., 2018). For example, Donkoh (1989) evaluated the differences in drinking water uptake  of broilers under four constant ambient temperatures: 20°, 25°, 30° and 35°C. Only the birds raised at 30° and 35°C showed a significant increase in water consumption.

In commercial broiler farms, the temperature is strictly controlled through ventilation. Therefore, major temperature deviations will be mitigated most of the time. There are exceptions of course ((sub)tropical climates and excessively hot summers), but these are of less importance in the target region of Western Europe.

Moreover, temperature-mediated alterations in drinking behaviour of the flock will not impact the administered dosage, since the concentration of the antimicrobial agent added to the drinking water is dependent on the water consumption of the previous 24 h. This can be appreciated from equation 1 in the paper.

However, for many antimicrobial agents and products, this dose calculation is not stated on the leaflet. Therefore, it can be predicted that concentration correction of the drug based on the water intake is not always done in practice. However, it can be considered good veterinary / agricultural practice to always determine the drug concentration that is needed to be added to the drinking water to attain a certain dose based on the average water consumption of the flock.

In our case (dosing based on water consumption), the shift in water uptake in the flock will be corrected for in the dose calculation. For this reason, an ambient temperature covariate that could influence the average water (and drug) uptake at different temperatures was not included in the DBPK model. However, future studies should be conducted to evaluate the impact of temperature deviations on drinking behaviour and subsequent absorption, disposition and efficacy of antimicrobial agents in more detail.

The main goal of this paper is to introduce a conceptual framework, namely the drinking behaviour PK (DBPK) model, for dose optimization of antimicrobial agents in poultry. Since this is a model, abstraction of reality is made and only the most important influencing factors (in our case the inter-individual variability in drinking water uptake) are taken into account. We hope that the reviewer concurs with this reasoning. To avoid any confusion and to let the readers know that this was taken into account, we have included a summary of our explanation in the paper (highlighted in yellow), see lines 201-225. Additionally, we also mentioned the temperature at which the animals were housed (lines 292-293 and 307-309).

References

Curtis SE. Environmental management in animal agriculture. Ames: Iowa State University Press (1983).

Donkoh A. Ambient temperature: a factor affecting performance and physiological response of broiler chickens. Int J Biometeorol (1989) 33:259–265. doi:10.1007/BF01051087

Mack LA, Felver-Gant JN, Dennis RL, Cheng HW. Genetic variations alter production and behavioral responses following heat stress in 2 strains of laying hens. Poult Sci (2013) 92:285–294. doi:10.3382/ps.2012-02589

Mascarenhas NMH, da Costa ANL, Pereira MLL, de Caldas ACA, Batista LF, Gonçalves EL. Thermal conditioning in the broiler production: Challenges and possibilities. J Anim Behav Biometeorol (2018) 6:52–55. doi:10.31893/2318-1265JABB.V6N2P52-55

May JD, Lott BD. Feed and water consumption patterns of broilers at high environmental temperatures. Poult Sci (1992) 71:331–336. doi:10.3382/ps.0710331

May JD, Lott BD, Simmons JD. Water Consumption by Broilers in High Cyclic Temperatures: Bell Versus Nipple Waterers. Poult Sci (1997) 76:944–947. doi:10.1093/ps/76.7.944

Round 2

Reviewer 2 Report

I respect and appreciate the authors' responses to my comments and concerns.  However, I still believe this is a bigger issue than even the few references they have provided are addressing.  In the US, which is the most familiar production practice of broilers I am familiar with, but I am told it is very similar to EU practices, it is a fully integrated system, where birds essentially are housed in the same barn from hatching to finishing.  Thus they are never moved.  They could be treated at any stage of their production lives.  The paper presented and the references provided appear to only deal with water (and thus drug consumption) in a very narrow range of that production cycle, which is near the end.  This does not help the young birds/chicks very much when you then add in the rapid body weight changes to these animals.  (R.P. Hunter, I. Mahmood, & M.N. Martinez (2008) Prediction of xenobiotic clearance in avian species using mammalian or avian data: How accurate is the prediction?  J. vet. Pharmacol. Therap., 31, 281-284.). 

While I support the need for this data and truly wish that it would be used, especially in the US, the reality is is that it won't.  In the US, the product isn't marketed or approved anymore. In addition, due to marketing pressure and cost of goods, it is cheaper to euthanize the flock when there is a disease outbreak than to treat with an antimicrobial.  This reviewer questions the animal welfare issues regarding this, but that is the economics of the business.  

So, for this reviewer, this paper has a VERY limited scope.  It addresses ONLY broilers at an age close to finish, not the entire production cycle for just an EU issue, as stated in the introduction.  For an international journal, such a narrowly defined scope is really not acceptable and recommend that this paper be submitted to a regional journal, such as British Poultry Science mainly due to this narrow scope and subject as it is not for the broad readership.

Author Response

We respect the reviewers decision to reject our paper. However, we’d like the reviewer and editor to consider the following aspects which, in our opinion, can address the reviewers remarks.

With respect to the reviewer’s initial comment (round 1) regarding the influence of temperature on the drinking behaviour and its impact on the dose optimization process, we believe we have provided substantial evidence validating the exclusion of an ambient temperature covariate that could influence the average water (and drug) uptake at different temperatures in the DBPK model (see round 1 response).

First, many studies show that drinking water uptake is only significantly increased in periods of heat stress. In commercial broiler farms, the temperature is strictly controlled through ventilation. Therefore, major temperature deviations will be mitigated most of the time. There are exceptions of course ((sub)tropical climates and excessively hot summers), but these are of less importance in the target region of Western Europe.

Second, temperature-mediated alterations in drinking behaviour of the flock will not impact the administered dosage, since the concentration of the antimicrobial agent added to the drinking water is dependent on the water consumption of the previous 24 h.

With regards to the comments put forward in the second, rejection report:

They could be treated at any stage of their production lives.  The paper presented and the references provided appear to only deal with water (and thus drug consumption) in a very narrow range of that production cycle, which is near the end.  This does not help the young birds/chicks very much when you then add in the rapid body weight changes to these animals.

The focus of our study was the dose optimization of enrofloxacin for the treatment of respiratory colibacillosis (and subsequent septicaemia) due to APEC infections. This syndrome mostly occurs in the in the second half of the production round (e.g. 3 - 4 weeks or older) (Nakamura el al., 1985; Kim et al., 2020). Therefore, for this indication, treatment in the first weeks of age of the birds is unlikely. Additionally, as stated further in the rebuttal, for infections during the first days of age, next to the treatment with antimicrobials, it is often more cost-effective to cull the diseased animals.

As mentioned in the paper, we have investigated the PK of enrofloxacin in three separate animal trails: one IV and PO trail on birds of 4 weeks of age, one PO trail where the animals were 27, 29, 34 or 36 days of age and a drinking water medication trail when the broilers were 5 weeks old. Thus, the application of our DBPK model lies in the second half of the production round, from approximately 4 weeks till the end (6 weeks). Considering the lifetime of a broiler, this timeframe is quite substantial (approximately 50%) and therefore not very narrow in our opinion.

Looking at the results of the paper, the difference in the PK parameters (e.g. clearance or volume of distribution) between birds of 4 or 5 weeks of age is minimal, indicating that most of the variability in PK is due to the drinking behaviour. Therefore, body weight changes in the second half of the broiler’s life play only a limited role in the PK of enrofloxacin. Also, differences in body weight between the different ages do not impact the dosing, since the dose is corrected based on the average weight of the animals (see equation 1 in paper, dose in mg/kg).

While I support the need for this data and truly wish that it would be used, especially in the US, the reality is is that it won't.  In the US, the product isn't marketed or approved anymore. In addition, due to marketing pressure and cost of goods, it is cheaper to euthanize the flock when there is a disease outbreak than to treat with an antimicrobial.  This reviewer questions the animal welfare issues regarding this, but that is the economics of the business.

Indeed, in the US the use of enrofloxacin has been banned since 2005 due to the development of fluoroquinolone resistance in Campylobacter strains of poultry origin (FDA, 2005). In the European Union (EU) however, risk evaluation did not result in the banning of the product, but stated that FQ should be reserved for the treatment of clinical conditions that have responded poorly to other classes of antimicrobials (EMA, 2006). More recently, as stated in the paper, in 2019, EMA/CVMP released a draft stating the urgency to optimize the dosage regimens of antimicrobial drugs currently employed in veterinary medicine, especially those administered via the drinking water, including enrofloxacin using modelling/simulation and probability of target attainment (PTA) analysis. This was the main reason to conduct this study. In summary, except in the US, FQ’s are still frequently used in the rest of the world (EU, Asia, Latin-America) (Joosten et al., 2019; Li et al., 2007; Sang et al., 2016; Morales-Barrera et al., 2016). Therefore, this paper addresses not only an EU issue, but a worldwide issue.

The reviewer also states that euthanizing the entire flock is more economical than treating with an antimicrobial agent. This only hold true when the birds are less than a week old (Christensen et al., 2021). Moreover, euthanizing an entire flock is not common practice, only the debilitated birds are removed. When the birds are older, a lot of investment has been made by the farmer (e.g. water, feed). Because of this, antimicrobial therapy will be initiated in those cases to try to save as many animals of the flock in order to achieve return on investment. Additionally, ethical considerations do play an important role in the decision to treat the animals with antimicrobials and cannot be as easily dismissed as stating that it is just the economics of the business.

So, for this reviewer, this paper has a VERY limited scope.  It addresses ONLY broilers at an age close to finish, not the entire production cycle for just an EU issue, as stated in the introduction.  For an international journal, such a narrowly defined scope is really not acceptable and recommend that this paper be submitted to a regional journal, such as British Poultry Science mainly due to this narrow scope and subject as it is not for the broad readership.

To conclude, we would like to the attention to the added value of this study. Our paper introduces the novel concept of the DBPK model and describes a novel approach for the PK modeling, dose optimization and clinical breakpoint determination of antimicrobial agents administered via the drinking water in the veterinary sector. The developed workflow of the DBPK model can also be applied to other antimicrobial drugs and even different types of medication. Dose optimization of currently approved antimicrobials in poultry will in turn contribute to the judicious use of antimicrobial agents and subsequently mitigate the development of antimicrobial resistance and improve the One Health sustainability.

References

Christensen H, Bachmeier J, Bisgaard M. New strategies to prevent and control avian pathogenic Escherichia coli (APEC). Avian Pathol (2021) doi:10.1080/03079457.2020.1845300

EMA. 2006. The European Agency for Evaluation of Medicinal Products. Reflection paper on the use of fluoroquinolones in food-producing animals in the European union: Development of resistance and impact on human and animal health. EMEA/ CVMP/SAGAM/184651/2005.

European Medicines Agency. Reflection paper on dose optimisation of established veterinary antibiotics in the context of SPC harmonisation. Comm Med Prod Vet Use (2019) 44:1–120. Available at: www.ema.europa.eu/contact [Accessed May 10, 2021]

FDA. 2005. U.S. Food and Drug Administration. Withdrawal of approval of the new animal drug application for enrofloxacin in poultry; Docket No. 2000N–1571. Accessed May 10, 2021. http:// www.fda.gov/AnimalVeterinary/SafetyHealth/RecallsWithdraw- als/ucm042004.htm.

Joosten P, Sarrazin S, Van Gompel L, Luiken REC, Mevius DJ, Wagenaar JA, Heederik DJJ, Dewulf J, Graveland H, Schmitt H, et al. Quantitative and qualitative analysis of antimicrobial usage at farm and flock level on 181 broiler farms in nine European countries. J Antimicrob Chemother (2019) 74:798–806. doi:10.1093/jac/dky498

Kim Y Bin, Yoon MY, Ha JS, Seo KW, Noh EB, Son SH, Lee YJ. Molecular characterization of avian pathogenic Escherichia coli from broiler chickens with colibacillosis. Poult Sci (2020) 99:1088–1095. doi:10.1016/j.psj.2019.10.047

Li Q, Bi X, Diao Y, Deng X. Mutant-prevention concentrations of enrofloxacin for Escherichia coli isolates from chickens. Am J Vet Res (2007) 68:812–815. doi:10.2460/ajvr.68.8.812

Morales-Barrera E, Calhoun N, Lobato-Tapia JL, Lucca V, Prado-Rebolledo O, Hernandez-Velasco X, Merino-Guzman R, Petrone-García VM, Latorre JD, Mahaffey BD, et al. Risks involved in the use of Enrofloxacin for Salmonella enteritidis or Salmonella heidelberg in commercial poultry. Front Vet Sci (2016) 3:72. doi:10.3389/fvets.2016.00072

Nakamura K, Maeda M, Imada Y, Imada T, Sato K. Pathology of spontaneous colibacillosis in a broiler flock. Vet Pathol (1985) 22:592–597. doi:10.1177/030098588502200614

Sang K, Hao H, Huang L, Wang X, Yuan Z. Pharmacokinetic–Pharmacodynamic Modeling of Enrofloxacin Against Escherichia coli in Broilers. Front Vet Sci (2016) 2:80. doi:10.3389/fvets.2015.00080